# Utility of In Vitro Cellular Models of Low-Dose Lipopolysaccharide in Elucidating the Mechanisms of Anti-Inflammatory and Wound-Healing-Promoting Effects of Lipopolysaccharide Administration In Vivo

**DOI:** 10.3390/ijms241814387

**Published:** 2023-09-21

**Authors:** Teruko Honda, Hiroyuki Inagawa

**Affiliations:** 1Department of Medical Technology, School of Life and Environmental Science, Azabu University, Sagamihara 252-5201, Japan; 2Research Institute for Healthy Living, Niigata University of Pharmacy and Applied Life Sciences, Niigata 956-0841, Japan; pina@shizenmeneki.org; 3Control of Innate Immunity, Collaborative Innovation Partnership, Takamatsu 761-0301, Japan

**Keywords:** lipopolysaccharide, monocyte, macrophage, inflammation, lifestyle-related diseases

## Abstract

Lipopolysaccharide (LPS) is a bacterial component that activates intracellular signaling pathways upon binding to the Toll-like receptor (TLR)-4/MD-2 complex. It is well known that LPS injected into animals and high-dose (100 ng/mL to 1 μg/mL) LPS treatment to innate immune cells induce an inflammatory response. In contrast, LPS is naturally present in the gastrointestinal tract, respiratory tract, and skin of humans and animals, and it has been shown that TLR-4-deficient animals cannot maintain their immune balance and gut homeostasis. LPS from commensal bacteria can help maintain homeostasis against mucosal stimulation in healthy individuals. Oral LPS administration has been shown to be effective in preventing allergic and lifestyle-related diseases. However, this effect was not observed after treatment with LPS at high doses. In mice, oral LPS administration resulted in the detection of LPS at a low concentration in the peritoneal fluid. Therefore, LPS administered at low and high doses have different effects. Moreover, the results of in vitro experiments using low-dose LPS may reflect the effects of oral LPS administration. This review summarizes the utility of in vitro models using cells stimulated with LPS at low concentrations (50 pg/mL to 50 ng/mL) in elucidating the mechanisms of oral LPS administration. Low-dose LPS administration has been demonstrated to suppress the upregulation of proinflammatory cytokines and promote wound healing, suggesting that LPS is a potential agent that can be used for the treatment and prevention of lifestyle-related diseases.

## 1. Introduction

Lipopolysaccharide (LPS) is a major component of the outer membrane of Gram-negative bacteria [1]. The binding of LPS to the Toll-like receptor (TLR)-4/MD-2 complex, activates the intracellular MyD88/NF-κB signaling pathway, leading to the production of proinflammatory cytokines, such as interleukin 1β (IL-1β), IL-6, and tumor necrosis factor-alpha (TNF-α), which induce inflammatory responses [2,3]. Intravascular or intraperitoneal administration of LPS has been shown to induce systemic production of these proinflammatory cytokines, leading to endotoxin shock, which includes hypotension and fever [4]. High-dose LPS administration (100 ng/mL to 1 µg/mL) is commonly used in vitro to mechanistically analyze sepsis and chronic inflammatory diseases (e.g., diabetes, dyslipidemia, and inflammatory bowel disease). Given that high-dose LPS treatment is a well-known inducer of inflammation, it has widely been used to promote inflammatory reactions (Figure 1).

LPS-producing bacteria are ubiquitous in the environment and account for approximately 50% of the >100 trillion symbiotic bacteria naturally present in the gastrointestinal tract, respiratory tract, and skin of healthy individuals. LPS is also found in several edible plants, such as wheat and rice, thereby being routinely ingested through food [5,6]. It has been reported that LPS intake from the environment and foods plays a role in maintaining health [7]. Reduced LPS intake during infancy has been associated with an increased risk of developing allergic diseases [8]. Furthermore, LPS has been reported to enhance TLR-4 expression in keratinocytes during skin wound healing and induce growth factors that promote wound healing [9]. TLR-4-deficient mice exhibit reduced peristaltic intestinal movement and are more likely to experience constipation [9,10]. Therefore, LPS intake from the environment plays a crucial role in maintaining immune balance and gut homeostasis. These functions of LPS as a modulator of the immune system and as a regulator of gut homeostasis differ from its harmful effects as an endotoxin or inflammation inducer. These findings suggest that intravascular and in vitro administration of LPS at high doses are suitable for examining inflammatory responses but not for investigating the physiological functions of LPS administered via oral and transdermal routes (Figure 1).

Although LPS is ubiquitously found in the environment and food, LPS concentrations to which cells are exposed through these sources are unknown. Intravenous administration of low-dose LPS to humans has been reported to induce an inflammatory response, with an MTD of 1~4 ng/kg [11]. However, environment- or food-derived LPS does not seem to induce inflammatory responses [8,12]. In addition, no toxicity or exacerbated inflammatory responses after oral and transdermal LPS administration have been observed in animal models [12,13]. This means that the response varies depending on the LPS dose and route of administration. Oral LPS administration is effective in preventing chronic inflammatory diseases (e.g., type II diabetes) [14,15] and cognitive decline and improving obesity and atherosclerosis [16,17]. In type II diabetes model mice (KK/Ay mice), oral LPS administration has been shown to improve insulin resistance and glucose intolerance and induce the expression of adiponectin (an anti-inflammatory cytokine) in the adipose tissue [15]. Orally administered LPS has been shown to reduce atherosclerosis plaque deposition in the aorta, improve glucose tolerance, and decrease low-density lipoprotein (LDL) in ApoE-deficient atherosclerosis mice [16]. Oral LPS administration has also been reported to prevent cognitive decline in the brain of a mouse model of diabetic dementia induced by intracerebroventricular administration of streptozotocin. It was also observed that oral LPS administration did not alter proinflammatory cytokine levels and exhibited neuroprotective effects through the activation of the anti-inflammatory cytokine IL-10 [18]. Oral LPS administration at a dose of 1 mg/kg body weight daily for 1 week resulted in the detection of LPS at a low concentration (8.5 pg/mL) in the peritoneal fluid but not in the blood. In contrast to intravascular LPS administration, oral LPS administration did not induce the production of the proinflammatory cytokines IL-1β, IL-6, and TNF-α [19]. These results suggest that LPS administered via the oral route may prevent chronic inflammatory diseases and cognitive decline and improve the state in animal models with obesity and atherosclerosis. The beneficial effects of oral LPS administration may be attributed to its ability to modulate anti-inflammatory responses and promote wound healing. However, the physiological functions of LPS and the potential mechanisms underlying the effects of oral and transdermal LPS administration are still not fully understood (Figure 1).

We hypothesize that the doses of LPS to which cells are exposed after oral and transdermal administration are lower than those generally used in vitro. Indeed, several studies have suggested that low-dose LPS has beneficial effects in vitro [20,21,22,23]. In in vivo experiments, it is difficult to observe the effects of only low-dose LPS-activated cells because of the network of different cell types involved; in vivo experiments may also reflect the effects of low-dose LPS in other cell types. In this work, we review recent evidence supporting the anti-inflammatory and wound-healing-promoting effects of low-dose LPS and provide information on the utility of in vitro experiments to elucidate the mechanisms.

## 2. Usefulness of Low-Dose LPS-Activated Monocyte-Lineage Cells as an In Vitro Experimental Model

Monocytes are versatile immune cells that sense environmental changes and differentiate into tissue-specific macrophages [24,25,26,27,28]. These cells (monocytes and macrophages) contribute to the maintenance of homeostasis through different signaling pathways. Monocytes and macrophages are also primary phagocytic cells of the immune system that play an important role in defense against pathogens and foreign substances [29,30]. In contrast, macrophages that accumulate in tumor or obese adipose tissues can increase the production of proinflammatory cytokines through cell-to-cell interactions. Macrophage accumulation in these tissues and subsequent increased production of proinflammatory cytokines cause inflammatory reactions and induce chronic inflammation [31,32], the two main hallmarks involved in the malignant transformation of cancer cells and the development of lifestyle-related diseases such as diabetes and atherosclerosis. As monocytes and macrophages express TLR-4, we performed in vitro experiments to activate these cells by LPS, which has been shown to exert anti-inflammatory effects when administered orally. Our studies revealed that low-dose LPS (100 pg/mL)-activated monocytes/macrophages exhibited decreased levels of proinflammatory cytokines, unlike high-dose LPS-activated monocytes/macrophages [20,21]. These results indicate that treatment with different doses of LPS elicits distinct inflammatory reactions.

It is known that when LPS binds to the TLR-4/MD-2 complex, it initiates intracellular signaling and activates NF-κB, a major inflammatory cascade. There are five types of proteins that constitute NF-κB (NF-κB transcription factor family): p50, p52, p65(RelA), c-Rel, and RelB. There may be differences in the signaling mechanisms of high-dose and low-dose LPS in the regulation of NF-κB. There are two main types of NF-κB activation: classical pathway and alternative pathway. In the former, when IκB bound to NF-κB is degraded by IκB kinase (IKK) complex (NEMO, IKKα, and IKKβ), NF-κB, which has been bound to IκB and inactivated, is activated and translocated into the nucleus to act as a transcription factor. In the latter, upon activation of NIK and subsequent activation of a complex composed of IKKα dimers, RelB/p100 undergoes limited degradation to become RelB/p52, which translocated into the nucleus to act as a transcription factor [33]. High-dose LPS activates the classical pathway and induces inflammatory cytokines. On the other hand, intracellular signaling pathways by low-dose LPS are not clear, although there are several reports on the regulation of NF-κB. Low-dose LPS-activated macrophages do not cause degradation of IκB and preferential reduction of RelB [19,34,35]. How anti-inflammation and wound healing are regulated by cytokine induction requires further study. As mentioned above, in vitro experiments suggest that NF-κB may be differentially regulated by LPS concentration (Figure 2).

LPS is a major constituent of the outer membrane of Gram-negative bacteria [1]. Commensal Gram-negative bacteria, which comprise approximately half of the microbiota in the gastrointestinal and respiratory tracts and skin, are a significant source of LPS. Oral and transdermal LPS administration does not elicit evident inflammatory reactions, whereas sepsis has been reported in inflammatory reactions induced by intravenous administration of LPS. However, the details of the physiological functions of LPS remain unclear. In vitro experiments may clarify the molecular mechanisms triggered by treatment with low-dose LPS, and monocyte/macrophage differentiation may provide evidence for understanding the beneficial effects of LPS under physiological conditions.

LPS is a potent inducer of inflammatory responses in vitro because high-dose LPS administration induces monocyte/macrophage activation, leading to increased levels of proinflammatory cytokines. In contrast to high-dose LPS, low-dose LPS (50 pg/mL) has been shown to suppress the expression of genes encoding proinflammatory (i.e., *IL6* and *TNFA*) but not anti-inflammatory (i.e., *IL10*) cytokines in mouse bone marrow-derived macrophages [36]. Furthermore, in human-derived monocyte-lineage cells (THP-1), treatment with low-dose LPS (100 pg/mL) has been shown to inhibit the upregulation of genes encoding the proinflammatory cytokines *IL1B* and *IL8* without altering the expression of the gene encoding the anti-inflammatory cytokine *IL10*. Thus, monocytes/macrophages activated with low-dose LPS show a phenotype different from those activated with high-dose LPS and exhibit anti-inflammatory effects similar to those observed after oral LPS administration.

Macrophage accumulation in tumor tissues has been associated with increased production of proinflammatory cytokines through cell-to-cell interactions, suggesting that macrophages contribute to the malignant transformation of cancer cells by inducing chronic inflammation in tumor tissues [37,38]. The use of cell culture inserts to coculture THP-1 and human-derived colon cancer cells (DLD-1) has provided a valuable model for studying the crosstalk between macrophages and cancer cells in tumor tissues. This model showed that THP-1 cells induce the expression of *IL1B* and *IL8* in this coculture using a cell culture insert. Conversely, low-dose LPS (100 pg/mL)-activated THP-1 has been shown to suppress the upregulation of the proinflammatory cytokines *IL1B* and *IL8* without altering the levels of the anti-inflammatory cytokines *IL10* and *TGFB1* [20,21]. Therefore, the anti-inflammatory effects of low-dose LPS-activated monocyte-lineage cells in vitro are similar to those observed after oral LPS administration.

The use of animals in research has come under increasing scrutiny in recent years, leading to the development of alternative experimental methods. Furthermore, the results of animal studies are not always predictive of the results of human clinical trials. Therefore, we consider that low-dose LPS-activated monocytes/macrophages may provide a valuable in vitro model for studying the effects and physiological relevance of oral LPS administration.

## 3. Regulation of Proinflammatory Cytokine Levels in Adipocytes and Vascular Endothelial Cells Using Low-Dose LPS-Activated Monocyte-Lineage Cells

Macrophage accumulation in the adipose tissues of patients with type II diabetes is a major driver of local chronic inflammation that contributes to the emergence of lifestyle-related diseases [31,32]. LPS has been suggested to exacerbate this chronic inflammation response. Conversely, oral LPS administration has been reported to have beneficial effects in KK/Ay mice by suppressing the expression of diabetes-associated indices, such as the oral glucose tolerance test, hemoglobinA1c (HbA1c), and glucose tolerance index (Homeostatic Model Assessment for Insulin Resistance, HOMA-IR), and increasing the expression of the anti-inflammatory cytokine adiponectin [15]. The vascular endothelial cells play a fundamental role in maintaining vascular homeostasis by exchanging substances between the blood and surrounding tissues and producing physiologically active substances. The elevated proinflammatory cytokines levels in the diabetes patient’s blood exacerbate vascular endothelial cell activation and promote blood coagulation. The excessive inflammatory response of vascular endothelial cells may promote the formation of thrombi, forming blood clots that block vessels in atherosclerosis. In ApoE-deficient mice, oral LPS administration has been shown to reduce atherosclerotic lesions and decrease lipid and proinflammatory marker levels [16]. These findings suggest that oral LPS administration exerts beneficial effects in adipocytes and vascular endothelial cells, although further in vitro studies are needed to confirm this hypothesis and elucidate the underlying mechanisms.

In a previous study, we showed that the addition of a conditioned medium of THP-1 treated with low-dose LPS (100 pg/mL) to human-derived adipocytes suppressed the upregulation of genes encoding the proinflammatory cytokines *IL6* and *IL8*, unlike the addition of a conditioned medium of THP-1 treated with high-dose LPS (1 mg/mL) (Table 1) [22]. Moreover, the addition of a conditioned medium of THP-1 treated with low-dose LPS (100 pg/mL) to human-derived vascular endothelial cells (HAoECs) can significantly suppress the upregulation of genes encoding inflammatory cytokines (*IL1B* and *IL8*) but not genes encoding anti-inflammatory cytokines (*TGFB1*) in HAoECs (Table 1) [23]. The anti-inflammatory effects of low-dose LPS-activated monocyte-lineage cells were demonstrated in vitro using adipocytes and HAoECs. These cells can effectively inhibit the upregulation of proinflammatory cytokine genes, suggesting a role in regulating the excessive inflammatory response in adipocytes and vascular endothelial cells. These in vitro experiments provide insights into the pathogenesis of lifestyle-related diseases characterized by chronic inflammation.

## 4. Regulation of Factors Involved in Lifestyle-Related Diseases by Low-Dose LPS

The plasminogen activator inhibitor 1 (PAI-1) factor, a serine protease inhibitor expressed in adipocytes, vascular endothelial cells, and other cell types, inhibits fibrinolysis by binding to and thus inactivating plasminogen. Visceral fat accumulation and inflammatory responses have been associated with increased PAI-1 expression, which may lead to thrombosis. Therefore, PAI-1 is considered a contributing factor to the development of atherosclerosis [39]. As shown in Table 1, the addition of a conditioned medium of THP-1 treated with low-dose LPS (100 pg/mL) to human-derived adipocytes significantly suppressed the upregulation of the *SERPINE1* [22]. The addition of a conditioned medium of THP-1 treated with low-dose LPS (100 pg/mL) to HAoECs has also been shown to inhibit the increased expression of *SERPINE1* (Table 1) [23]. The results of in vitro experiments using adipocytes and vascular endothelial cells, which reflect the crosstalk with monocytes/macrophages, revealed that these cells inhibited the upregulation of the *PAI-1* gene through the anti-inflammatory effects of low-dose LPS-activated monocyte-lineage cells.

Adiponectin is an adipokine secreted by adipocytes that activate AMP-activated protein kinase (AMPK) and peroxisome proliferator-activated receptor (PPAR)α via the adiponectin receptors (Adipor1 and Adipor2) [15]. Activation of these factors promotes fatty acid burning and sugar uptake, decreases triglyceride content, and improves insulin resistance. It has been shown that patients with diabetes and atherosclerosis have low adiponectin levels in the blood. Moreover, it has been reported that adiponectin concentration in the blood is inversely correlated with visceral fat mass [39]. Therefore, adiponectin may play a role in lifestyle-related diseases, including diabetes and atherosclerosis [40].

The expression of the gene encoding adiponectin (*ADIPOQ*) is significantly increased in human-derived adipocytes upon the addition of a conditioned medium of THP-1 treated with low-dose LPS (100 pg/mL) (Table 1) [22]. It has been reported that the expression of *Adipor1* and *Adipor2* in adipose tissues of KK-Ay mice was increased in the oral LPS administration group compared with those in the control group [15]. Therefore, oral LPS administration was shown to suppress glucose intolerance and insulin resistance. It is suggested that low-dose LPS may help prevent lifestyle-related diseases by upregulating adiponectin expression.

## 5. Therapeutic Efficacy of Low-Dose LPS-Activated Cardiomyocytes in Myocardial Ischemia-Reperfusion Injury

Apoptosis is a cellular mechanism to eliminate defective cells in multicellular organisms. The death of heart muscle cells by programmed cell death, frequently referred to as “myocardial apoptosis” is a common feature of myocardial infarction and other cardiac diseases. The apoptotic dead cells are then quickly eliminated by phagocytic cells (e.g., macrophages). The phagocytosis of apoptotic myocardial cells helps suppress the inflammatory response caused by the leakage of cellular contents from dead cells. Therefore, myocardial apoptosis regulates disease progression in myocardial infarction and other cardiac diseases.

In a rat model of myocardial ischemia–reperfusion (I/R) injury, intravenous high-dose LPS administrated one hour before ischemia has been shown to impair myocardial function; however, low-dose LPS treatment did not affect myocardial function [41]. Moreover, low-dose LPS (50 ng/mL) treatment suppressed cardiomyocyte apoptosis and reduced myocardial infarction size in rat cardiomyocyte H9C2 cells [42]. Caspase 3 is an important factor in apoptosis. Caspase 3 cleaves B-cell lymphoma-2 (Bcl-2) and Bcl-extra large (X_L_), causing them to lose their anti-apoptotic function and release a C-terminal fragment that promotes apoptosis. The expression of caspase-3 was also suppressed in low-dose LPS (50 ng/mL)-activated H9C2 cells [42]. In myocardial I/R injury, the levels of inositol requiring 1 (IRE1) signaling pathway-related factors Grp78, IRE1, p-ASK1, ASK1, p-JNK, and JNK were significantly increased at both the gene and protein levels. In contrast, low-dose LPS (50 ng/mL) significantly decreased the expression of these factors in H9C2 cells [42]. The results of in vitro experiments using LPS-activated cardiomyocytes suggest that low-dose LPS administration may have cardioprotective effects by inhibiting apoptosis and suppressing the expression of IRE1 signaling pathway-related factors.

In vivo experiments with high-dose LPS showed to be harmful to myocardial function and exacerbate I/R injury, whereas in vivo low-dose LPS administration did not show significant effects on myocardial function. In vitro experiments with low-dose LPS have been shown to inhibit cardiomyocyte apoptosis, suppress the expression of IRE1 signaling pathway-related factors, and reduce myocardial infarction size. Thus, in both in vivo and in vitro experiments, low-dose LPS treatment was found to have no impairing effects on myocardial function and to be useful in the treatment of myocardial I/R injury, in contrast to high-dose LPS administration. These results suggest that the dose of LPS is critical in determining its effects on myocardial function and I/R injury.

## 6. Beneficial Effects of Low-Dose LPS Treatment on Neurons Cells in Spinal Cord Injury

Spinal cord injury (SCI) is a serious trauma to the central nervous system. Previous reports revealed that low-dose LPS can significantly improve the recovery of motor function in a rat model of SCI [43]. Low-dose LPS administration has also been shown to have protective effects against SCI by decreasing caspase-3 levels and increasing the expression of Bcl-2; however, the associated mechanism remains to be elucidated. The treatment of PC12 neuron cells with high-dose LPS (5 µg/mL) has been shown to induce the release of the proinflammatory cytokines IL-1β, IL-6, and TNF-α in an in vitro model of SCI [44]. Moreover, low-dose LPS has been shown to increase the levels of Nrf2, p-PI3K/PI3K, and p-AKT/AKT and facilitate the nuclear translocation of Nrf2. In PC12 cells, low-dose LPS has been shown to inhibit apoptosis, as evidenced by the downregulation of pro-apoptotic caspase 3 and caspase 9 and the upregulation of anti-apoptotic HO-1, NQO1, and γ-GCS proteins. Low-dose LPS treatment has also been shown to reduce the apoptotic rate and decrease oxidative stress levels by activating the PI3K-AKT-Nrf2 signaling pathway in an in vitro PC12 cell model of SCI [45]. These results highlight the therapeutic potential of low-dose LPS administration in SCI, although further studies are needed to elucidate the underlying mechanisms beyond beneficial effects.

## 7. Limitations and Development of In Vitro Experiments Using Low-Dose LPS to Reflect Oral LPS Administration

Monocytes/macrophages play a key role in maintaining homeostasis by sensing environmental changes and adjusting cellular responses. In mice, oral LPS administration can be evidenced by detecting LPS at markedly low concentrations in the peritoneal fluid [19]. This finding suggests that the functional changes in low-dose LPS-activated monocyte/macrophage are similar to those elicited by oral LPS administration. The in vitro cell culture models used in these studies do not fully recapitulate the complex cellular interactions occurring in vivo. Therefore, it is important to be aware of the limitations of these models when interpreting the results.

Alzheimer’s disease is characterized by the deposition of the amyloid-β protein in the brain. Microglia are resident immune cells of the brain involved in clearing cellular debris and pathogens. However, in Alzheimer’s disease, microglial phagocytosis is reduced, leading to the accumulation of amyloid-β in the brain and the progression of the disease. In mice microglia, it has been shown that oral LPS administration does not induce the expression of IL-1β, IL-6, TNF-α, nitric oxide synthase 2 (NOS2) proinflammatory mediators and neither suppresses the level of the anti-inflammatory mediator peroxisome proliferator-activated receptor gamma (PPARγ) [17]. These results suggest that oral LPS administration may have the potential to treat diabetes-related cognitive dysfunction (DRCD) by converting microglia into cells with anti-inflammatory functions. In addition to their phagocytic functions, microglia also produce a variety of anti-inflammatory cytokines and chemokines. After oral LPS administration, healthy microglia showed significantly increased expression of colony-stimulating factor 1 (CSF1) receptor, IL-10, IL-12B, prostaglandin-E2 EP4 receptor, c-Jun, and heat shock protein family genes than DRCD-associated microglia. In addition, oral LPS administration led to higher levels of IL-12B, prostaglandin-E2 EP4 receptor, and HSPb1 genes in DRCD-developed mice than in the microglia of healthy mice [18,46]. However, microglia levels of proinflammatory mediators did not significantly change between orally administered LPS and DRCD-developed mice. These results indicate that oral administration of LPS alters the microglial function and that gene expression differs from that in DRCD model mice and healthy mice.

The mechanism of microglial activation induced by oral LPS treatment has already been investigated. Oral LPS administration induces the expression of membrane-bound CSF1 in leukocytes through the intestinal mucosa. CSF1 is a hematopoietic factor that promotes the differentiation of hematopoietic stem cells into monocytes. The migration of leukocytes to the brain and the subsequent signaling through CSF1 receptors on microglia are essential for the production of anti-inflammatory and neuroprotective factors. The in vivo system is more complex than the in vitro system, as it involves a network of different cell types, including leukocytes, microglia, and other immune cells. This makes it difficult to isolate the effects of low-dose LPS on microglia, as the effects of LPS on other cell types may also contribute to the observed outcomes [7].

Recent in vitro studies with low-dose LPS stimulation have shown similar results to those with oral LPS administration. Oral LPS administration induces the insulin signaling pathway in the adipose tissue of KK/Ay mice [15]. In vitro LPS administration at 1 ng/mL and 100 ng/mL has been shown to induce insulin signaling in 3T3-L1 adipocytes [47]. In primary peritoneal tissue-resident macrophages, three repeated treatments with low-dose LPS (1 ng/mL) had anti-inflammatory and neuroprotective effects similar to those by oral LPS administration [48]. Low-dose LPS (1 ng/mL)-stimulated C8-B4 microglial cells showed increased levels of both proinflammatory (e.g., IL1β, TNFA, and IL6) and anti-inflammatory (IL10) cytokines. Moreover, the repeated treatment for 3 consecutive days with low-dose LPS did not alter the expression of inflammatory cytokines but did increase the levels of the anti-inflammatory cytokines IL10 and Arg1. Both single and three low-dose LPS stimulation cycles did not change the phagocytic ability of monocytes/macrophages, which is required to maintain homeostasis [49]. Therefore, in vitro experiments involving the administration of LPS at low doses may be a valuable tool in anti-inflammatory and wound-healing activity studies. Future research is needed to elucidate the underlying mechanisms and determine the clinical significance of these findings.

## 8. Conclusions

In this review, we discussed the current evidence regarding low-dose LPS delivery. In contrast to harmful high-dose effects, LPS at low concentrations has been shown to exert anti-inflammatory effects, display therapeutic efficacy for treating myocardial I/R injury, and have protective properties on neuron cells after spinal cord damage. Low-dose LPS-induced cell activation may also be a major molecular determinant of intracellular signaling pathways. Although LPS has been used in many analytical models to study the inflammatory response, it should be noted that this immune reaction is triggered by high-dose LPS treatment. Therefore, the concept that LPS induces inflammation should not be universalized. Anti-inflammatory effects and markedly low concentrations of LPS in the peritoneal fluid have been reported upon oral LPS administration. The mechanism by which LPS activates cells is not yet fully understood. Recent studies have revealed that the pathway for microglial activation via oral LPS administration involves membrane-bound CSF1 cell-mediated signaling. Future studies are needed to clarify the mechanism of action of low-dose LPS in vivo. In vitro experiments using low-dose LPS provide insights into this mechanism. Furthermore, low-dose LPS has shown potential for the prevention of lifestyle-related and other diseases, encouraging the development of new therapeutic strategies.

## Figures and Tables

**Figure 1 ijms-24-14387-f001:**
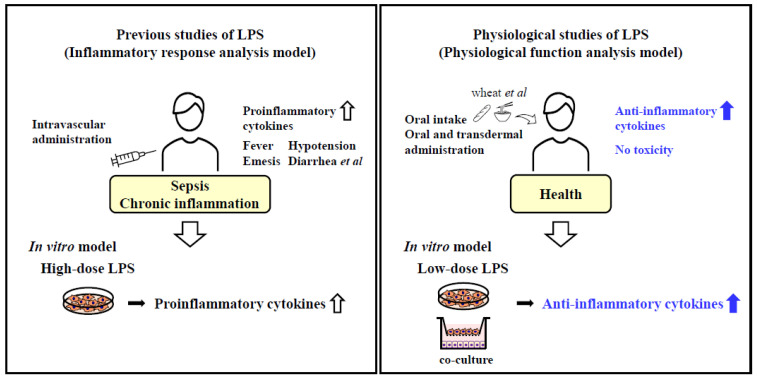
General studies (inflammatory response analysis model) and physiological studies (physiological function analysis model) of lipopolysaccharide (LPS). Intravascular administration of LPS is a model for sepsis and chronic inflammatory diseases, which increase the production of proinflammatory cytokines. In general studies, high-dose LPS has been used in in vitro inflammatory response analysis models. On the other hand, oral and transdermal administration of LPS is a model to evaluate the functionality of foods consumed daily such as brown rice, wheat bran, and buckwheat, which increase the production of anti-inflammatory cytokines as presented in this review. In physiological studies, low-dose LPS has been used in in vitro physiological function analysis models.

**Figure 2 ijms-24-14387-f002:**
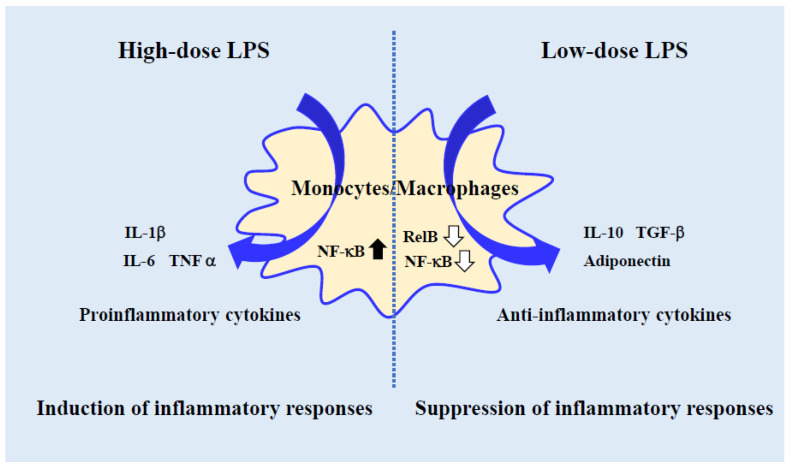
Signaling in monocytes/macrophages differs with LPS dose. High-dose LPS-activated monocytes/macrophages activate the NF-κB pathway and induce inflammatory responses. Whereas low-dose LPS-activated monocytes/macrophages inactive the NF-κB pathway and suppress inflammatory responses. The response of signaling pathways in LPS-activated monocytes/macrophages differs with LPS dose.

**Table 1 ijms-24-14387-t001:** Gene expression in Adipocytes and HAoECs through LPS-activated macrophages. The mRNA expression was analyzed using quantitative real-time PCR. Relative quantification was performed by normalizing the target expression to that of *ACTB*.

Adipocytes		**LPS**	**-**	**100 pg/mL**	**10 ng/mL**	**1 μg/mL**
	**Gene**					
	*IL6*		1.00	0.64	0.74	1.33
	*IL8*		1.00	0.50	0.57	1.81
	*SERPINE1*		1.00	0.82 *	0.77	1.06
	*ADIPOQ*		1.00	1.71 *	1.16	0.77
HAoECs		**LPS**	**-**	**100 pg/mL**	**10 ng/mL**	**1 μg/mL**
	**Gene**					
	*IL1B*		1.00	0.76 *	1.44	2.34
	*IL8*		1.00	0.57 *	1.66	1.85
	*TGFB1*		1.00	1.14	2.18	2.32
	*SERPINE1*		1.00	0.84 *	0.96	1.04

* *p* < 0.05.

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
