# Peer review of "Utility of In Vitro Cellular Models of Low-Dose Lipopolysaccharide in Elucidating the Mechanisms of Anti-Inflammatory and Wound-Healing-Promoting Effects of Lipopolysaccharide Administration In Vivo"

_ijms, 2023, doi:10.3390/ijms241814387_

Round 1

Reviewer 1 Report

The manuscript read well, I would suggest a description (or discussion) about the downstream molecues difference (the molecular mechanisms) of low-dose compared high-dose of LPS in monocytes/macriphages, not only focusing on phenotype summary in different cell types. This will provide the direction of this field studies.

Author Response

Dear Reviewer

Reviewer 2 Report

The review lacks important literature that shows that low doses of LPS can induce inflammatory response. Moreover, the context of some paragraphs in the review is not relevant to the general title .  As examples are the paragraphs entitled:

-5. Therapeutic efficacy of low-dose LPS-activated cardiomyocytes in myocardial ischemia-reperfusion injury

-6. Beneficial effects of low-dose LPS treatment on neurons cells in spinal cord injury

It would be better if it is restructured/rewritten and give a different title/focus of the review eg only in vivo conditions.

The quality of english language is good.

Author Response

Dear Reviewer

Reviewer 3 Report

This review manuscript by Honda and Inagawa concerns an important area of research highlighting the current status with the treatment with low-dose and high dose of LPS. In this review literature-based evidence is presented to show that low vs high dosage elicit different biological responses. Authors specifically concentrate on low-dosage of LPS could exert anti-inflammatory effects. However, at present molecular basis of such beneficial effects are not fully understood although it could be via CSP1 cell-mediated signalling. Overall this is an important area of research as most of the literature covers usually LPS-mediated proinflammatory response. 

Main concerns: 

1.     Authors should cover in more detail molecular details about regulation of expression of ADIPOQ.

2.     Similarly molecular details about down-regulation of caspase-3 levels. 

3.     Lines 31 and 133: It is more appropriate to use Gram-Negative instead of gram-negative. Also reference number 1 does not cover LPS its biosynthesis, role in bacterial physiology and structural aspects. Thus remove reference number 1 and replace by reviews on LPS such as  : Lipopolysaccharide endotoxins. Annu. Rev. Biochem. 200271, 635–700 and more recent review: Int. J. Mol. Sci. 2022, 23(1), 189.

Author Response

Dear Reviewer

Round 2

Reviewer 2 Report

The authors changed the title to 'in vitro models..', but now the content does not match because there is a lot of information referring to in vivo models.

In their response to my initial comment about lack of literature they write the following: 'We would also be very grateful if you could advise us on specific references, as we would like to consult the literature that you consider important'. It is not my responsibility to suggest literature. This is actually the meaning of writing a review

Author Response

Dear Reviewer

Thank you for your critical review and comments on our Manuscript ID (ijms-2578905). We have responded to the comments of the reviewer’s and revised.

Thank you in advance for reviewing the revised manuscript.

Sincerely,
